# Exploring the In Vitro Effects of Cassava Diets and Enterococcus Strains on Rumen Fermentation, Gas Production, and Cyanide Concentrations

**DOI:** 10.3390/ani14223269

**Published:** 2024-11-13

**Authors:** Saowalak Lukbun, Chanon Suntara, Rittikeard Prachumchai, Waroon Khota, Anusorn Cherdthong

**Affiliations:** 1Tropical Feed Resources Research and Development Center (TROFREC), Department of Animal Science, Faculty of Agriculture, Khon Kaen University, Khon Kaen 40002, Thailand; saowalaklukbun@kkumail.com (S.L.); chansun@kku.ac.th (C.S.); 2Department of Animal Science, Faculty of Agricultural Technology, Rajamangala, University of Technology Thanyaburi, Pathum Thani 12130, Thailand; rittikeard_p@rmutt.ac.th; 3Department of Animal Science, Faculty of Natural Resources, Rajamangala University of Technology Isan, Sakon Nakhon Campus, Sakon Nakhon 47160, Thailand; waroon.kh@rmuti.ac.th

**Keywords:** cattle nutrition, cyanide-utilizing bacteria, degradation efficiency, feeding regime, nutritional components

## Abstract

Cassava roots or leaves serve as a cost-effective source of energy and protein for ruminants but contain high levels of hydrocyanic acid (HCN), which pose toxicity risks. Cyanide-utilizing bacteria (CUB), specifically *Enterococcus faecium* KKU-BF7 and *Enterococcus gallinarum* KKU-BC15, found in rumen fluid, have demonstrated the capability to reduce HCN levels. This study investigated the effects of adding CUB to cassava-based diets on HCN reduction, gas production, and in vitro digestibility. The results highlighted that supplementing cassava diets with *E. faecium* KKU-BF7, particularly at a HCN concentration of 600 mg/kg, significantly improved HCN degradation, cumulative gas production, in vitro digestibility, and volatile fatty acid (VFA) concentration. These findings underscore the potential of CUB in mitigating HCN toxicity and enhancing feed quality, thereby promoting sustainable livestock production and animal health.

## 1. Introduction

In tropical regions, fresh cassava roots and leaves provide an affordable source of energy and protein for ruminants, positioning them as a viable alternative to cassava chips. This accessibility, along with the ease of local cultivation, makes cassava a popular feed choice among farmers [1,2]. Introducing fresh cassava diets as a feed alternative may effectively improve the use of cassava feeds without requiring preprocessing [3,4]. Fresh cassava can be a valuable energy source in ruminant diets, but its use is limited by the presence of hydrocyanic acid (HCN), which poses a risk of prussic acid toxicity or hydrocyanic poisoning in animals [5,6]. Toxicity risks from cassava depend on both the amount consumed and the animal’s physiological condition [7]. Poisoning is typically triggered when animals consume excessive quantities, particularly in cases where cassava constitutes a large part of the diet or is fed without processing. Ingesting feeds with over 500 mg HCN/kg on a dry-weight basis is considered dangerous, as noted by Aminlari et al. [8]. Fresh cassava roots have been found to contain approximately 114 mg/kg of HCN on a fresh weight basis, which, if unprocessed, can contribute to significant HCN exposure in livestock diets [9]. Prolonged exposure can impair animal health, manifesting as respiratory distress, reduced growth performance, or even mortality in severe cases.

Reducing HCN concentrations in cassava feed is critical for ensuring its safe inclusion in ruminant diets, as high levels of HCN pose toxicity risks to these animals. However, ruminants possess a natural capacity to detoxify HCN through the action of enzymes present in both rumen microorganisms and animal tissues [10]. This detoxification mechanism mitigates the potential toxicity of HCN, allowing ruminants to tolerate cassava feed when consumed under controlled levels. Ingested HCN is rapidly absorbed, with toxicity symptoms appearing within 15 min when large quantities are ingested, suggesting that absorption primarily occurs in the rumen [8,11]. The degradation rate of 60% after 4 h still allows significant HCN absorption, highlighting the importance of rapid detoxification mechanisms. Rhodanese and β-mercaptopyruvate sulfurtransferase are key enzymes involved in this process [10,12]. Rhodanese catalyzes the transfer of sulfur to cyanide (HCN), converting it into thiocyanate (SCN), a compound significantly less toxic than HCN, which is subsequently excreted in the urine. This reaction is crucial as it reduces the systemic toxicity of HCN, enabling ruminants to handle cyanogenic feeds better. β-mercaptopyruvate sulfurtransferase works similarly by transferring sulfur from sulfur-containing molecules to HCN, facilitating its conversion to thiocyanate [11]. These enzymes are not only active in rumen microorganisms but are also present in the animal’s liver and kidneys, where rhodanese activity is particularly high, ensuring detoxification occurs both in the digestive system and within body tissues [9,10]. Through the combined activity of these enzymes, ruminants convert approximately 65–80% of ingested HCN into thiocyanate, thereby reducing HCN levels to non-toxic concentrations [13].

The process requires adequate sulfur for optimal function, and the addition of sulfur to the diet has been shown to enhance HCN detoxification. For instance, the inclusion of elemental sulfur with cassava-based feeds can neutralize HCN, as demonstrated by Cherdthong et al. [3]. However, excessive sulfur intake can lead to adverse effects, such as reduced feed consumption and polioencephalomalacia [10,11]. Furthermore, the rumen pH and forage-to-concentrate ratio also influence HCN degradation. A rumen pH below 6.0, typically associated with high-concentrate diets, can impair microbial activity and slow HCN degradation [14]. Therefore, understanding the balance of dietary components is crucial in optimizing HCN detoxification and minimizing toxicity risks in ruminants [15,16].

Biotechnology could be employed to incorporate innovative feeding techniques that reduce the toxicity of HCN in fresh cassava roots. In support of this, Sumadong et al. [17] found that rumen bacteria activate rhodanese and β-mercaptopyruvate sulfurtransferases, enabling ruminants to detoxify small amounts of HCN efficiently. The utilization of enzymes and microorganisms in technology is intriguing. According to Bhalla et al. [18], five enzymes, namely cyanide hydratase, cyanide dehydratase, cyanide monooxygenase, and rhodanese, have been identified for the purpose of reducing HCN poisoning. The thiosulfate receptor binds with HCN at an atomic level. This results in the formation of a thiocyanate compound, which has the ability to decrease its level of toxicity [7,19]. Rhodanese can be synthesized by various bacteria, including *Thiobacillus denitrificans* [20], *Pseudomonas aeruginosa*, *Pseudomonas fluorescens*, *Escherichia coli* [21], *Thiobacillus neopolitanus* [22], *Bacillus subtilis* [23], and *Fusarium solani* [24]. A recent study conducted by Khota et al. [4] has revealed the presence of cyanide-utilizing bacteria (CUB), i.e., *E. faecium* KKU-BF7 and *E. gallinarum* KKU-BC15, in rumen fluid. A reduction in HCN levels is the result of the efficient utilization of HCN as a nitrogen source for microbial growth by these bacteria. The CUB group exhibits a significant resistance to a diverse array of HCN molecules, potentially surpassing other bacteria in its ability to metabolize HCN [10].

The hypothesis was that supplementing fresh cassava root or leaf with cyanide-utilizing ruminal bacteria (*E. faecium* KKU-BF7 and *E. gallinarum* KKU-BC15) would reduce HCN levels while improving nutrient digestibility and fermentation characteristics in the rumen. The aim of this study was to investigate the effects of these bacterial supplements on HCN reduction, gas kinetics, and in vitro digestibility using gas production techniques.

## 2. Materials and Methods

### 2.1. Treatments and Experimental Design

The experiment followed a complete randomized design (CRD) and had a 2 × 2 × 3 (+1) factorial structure. Factor A was the sources of HCN, which included fresh cassava root and fresh cassava leaves. Factor B represented the concentrations of HCN at 300 and 600 parts per million (ppm) in fresh samples. The HCN levels of 300 and 600 mg/kg DM were selected based on the findings of Supapong and Cherdtong [14], where these levels, combined with rhodanese enzyme at 1 mg per 10^4^ ppm HCN, demonstrated safe HCN detoxification and improved feed utilization in Thai native beef cattle. Factor C involved the introduction of different types of cyanide-utilizing bacteria (CUB) inoculants, including no CUB, *E. faecium* KKU-BF7, and *E. gallinarum* KKU-BC15, at a concentration of 10^8^ FU/mL. The (+1) treatment in the test bottle did not contain any additional elements.

### 2.2. Cyanide-Utilizing Bacteria in Rumen Culture

The CUB was derived through the process of screening and isolation in the rumen of swamp buffalo and Thai native beef cattle, as documented in a study conducted by Khota et al. [4]. The isolated strains, identified as KKU-BF7, were found to be closely related to the species *E. faecium*, as labeled by KKU-BF7. Similarly, KKU-BC15 was identified as *E. gallinarum*, as labeled by KKU-BC15. The bacteria of interest were selected by enriching them in nutrient broth (Difco Laboratories, Detroit, MI, USA) using the method described by Moradkhani et al. [21]. Following a series of culture enrichments in a mineral medium incubation, the growing culture was diluted in a solution containing 8.5 g per liter of NaCl at dilutions ranging from 10^−1^ to 10^−5^. Lactobacilli MRS broth medium (Difco Laboratories, Detroit, MI, USA) was used to cultivate CUB for 24 h. A concentration of approximately 10^8^ CFU/mL was determined by measuring the absorbance at 660 nm [21]. The bacteria were then prepared for use in an in vitro investigation.

### 2.3. The Preparation of Rumen Inoculums and Animals

Two male Thai cattle, each weighing around 450 ± 30 kg, were used to collect rumen fluid. The bulls were kept in separate cages. The concentrate diet had a crude protein (CP) content of 18% and a total digestible nutrient (TDN) content of 75%. The meal was divided into two equal portions and fed to the cattle at around 08:00 h and 17:00 h, with each portion amounting to 0.5% of body weight (BW). Rice straw was constantly available. Mineral blocks and water were freely available. In the early morning, approximately one hour prior to feeding, the cattle’s ruminal fluid was collected. Multiple layers of cheesecloth were employed to filter the fluid, which was subsequently transferred to heated thermos flasks for transport to the laboratory. Artificial saliva was synthesized utilizing the methodology proposed by Menke and Steingass [25]. By combining synthetic saliva and rumen fluid in a 2:1 ratio, a composite rumen inoculum was produced. To remove the oxygen, CO_2_ was added to the serum bottles before submerging them in a water bath heated to 39 °C. Forty milliliters of the rumen solution was injected into the bottles by force one hour before being used as inoculum.

### 2.4. Substrates

The Kasetsart 50 variety of fresh cassava, which included both the root and the leaf, was gathered from a local market in Khon Kaen, Thailand. One-year-old cassava was freshly picked, cleaned, and cut into pieces that were passed through a 1 mm sieve. The cassava was prepared to achieve HCN concentrations of either 300 or 600 mg/kg DM, corresponding to fresh weights of 1.08 and 2.16 g. A substrate consisting of a mixture of rice straw and concentrate (in a ratio of 60:40) was utilized, with a total weight of 0.5 g placed in the bottles. Laboratory analytical procedures were performed on the concentrate, straw, and HCN sources to ensure accurate composition and appropriate concentration levels. In order to achieve a particle size that was able to pass through a filter with a pore size of one millimeter, the fresh cassava, rice straw, and concentrate diet samples were precisely pulverized. These ground samples were then prepared for the investigation of their nutritional composition and for the gas production study. The CUB cultures were each put into the bottom at a concentration of 10^8^ CFU/mL. Table 1 provides an inventory of the diets that were used as substrates in the experiment, including their contents and their chemical composition.

The temperature in the incubator was set to 39 °C so that the bottles could be used for an in vitro gas test. The caps were sealed with rubber and metal caps. Every 2 h, the bottles were stirred during each sampling step. The cumulative assessment of gas output was conducted through three distinct runs, each consisting of 3 replicates [(3 bottles per treatment × 13 treatments) + 5 bottles of blank)] × 3 runs. To establish baseline measurements, a total of five control bottles containing only ruminal inoculum were used as blanks. These blanks were prepared in the same manner as the experimental bottles but without the addition of any substrates. The mean gas output values from these control bottles served as a reference point. The net gas output for each experimental bottle was determined by subtracting the mean values of the blanks from the corresponding measured values. A total of 78 bottles were used for the experiment, with 3 bottles per treatment and 13 treatments. Two sampling durations, 4 and 8 h of incubation, were considered. Ruminal pH, ammonia–nitrogen (NH_3_-N), and VFA were measured independently for each bottle. An assessment of digestibility was conducted using a distinct set of 78 bottles (3 bottles per treatment × 13 treatments × 2 sample times at 12 and 24 h incubation). A total of 130 bottles were used to prepare the HCN concentration analysis. Each treatment consisted of 2 bottles, and there were 13 treatments in total. The sampling was performed at 0, 4, 8, 12 and 24 h of incubation, resulting in 5 sample times.

### 2.5. Analysis of Chemical Composition in Diets

Table 1 displays the samples of rice straw, concentrate, fresh cassava root, and fresh cassava leaves that underwent chemical analysis. The DM, ash, CP, and organic matter (OM) measurements were carried out in compliance with the recommendations provided by the AOAC [26]. The methodology described by Van Soest et al. [27] was used to analyze the neutral detergent fiber (NDF) and acid detergent fiber (ADF). The cumulative assessment of gas output was monitored at various incubation times, specifically at 0, 1, 2, 4, 6, 8, 10, 12, 18, 24, 36, 48, 72 and 96 h. The gas production data were analyzed using the Ørskov and McDonald [28] model, which can be expressed as follows:Y = a + b (1 − e^−ct^)
where Y = gas produced at time “t” (mL), a = gas production from the immediately soluble fraction (mL), b = gas production from the insoluble fraction (mL), c = gas production rate constant for the insoluble fraction (mL/h), t = incubation time (h), and a + b = the potential extent of gas production (mL).

After the incubation period, fermentation fluid was collected at 4 and 8 h. The purpose of this was to measure the ruminal pH. The collected liquor was then filtered using four layers of cheesecloth. The fermentation liquors were subjected to centrifugation at a force of 16,000 times the acceleration due to gravity for a duration of 15 min at a temperature of 4 °C. This process was carried out in order to collect the liquid portion above the sediment, which was then used for the determination of NH_3_-N [26]. Additionally, the VFA content was analyzed using gas chromatography. The gas chromatograph used for this analysis was the Wilmington, DE 5890A Series II model (Agilent Technologies, Inc, CA, United States), and a glass column measuring 180 cm in length and 4 mm in diameter was employed. A combination of 100 g/L SP-1200 and 10 g/L H_3_PO_4_ were packed into the column using 80/100 mesh Chromosorb WAW, which was provided by Supelco, a company based in Bellefonte, PA, USA. Following 12 and 24 h of incubation, the in vitro digestibility of DM (IVDMD) and OM (IVOMD) was assessed. The resulting content was then used to determine the in vitro NDF degradability (IVNDFD) and in vitro ADF degradability (IVADFD) using an Ankom filter bag [4].

A modified version of Fisher and Brown’s [29] procedure, notably the picric acid method, was used to quantify the amounts of HCN in the fermentation liquid at various time points (0, 4, 8 and 12 h of incubation). Standard potassium cyanide (KCN) solutions were used to create a linear calibration curve. A solution that contained 0.5% (*w*/*v*) picric acid and 0.25 M Na_2_CO_3_ was added to 0.05 mL parts of KCN solutions. This resulted in the achievement of the desired result. The KCN solutions were centrifuged at 15,000× *g* for 10 min at 4 °C before the aliquots were added. The resulting solutions were heated to boiling for a duration of 5 min, then diluted to a volume of 1 mL with 0.85 mL of distilled water and cooled for a period of 30 min in tap water. Absorbance at 520 nm was measured using a spectrophotometer, with distilled water and picric acid reagent serving as the blank. For the purpose of calculating the degrading efficiency (*DE*) of HCN, the following formula was utilized:$$DE\;\left(\%\right)=\left(\frac{Ic-Rc}{Ic}\;\right)\times 100$$
where *Ic* = initial concentration of HCN (ppm) and *Rc* = residual concentration of HCN (ppm).

### 2.6. Statistical Analysis

Statistical analysis was conducted using the PROC GLM procedure in the SAS program version no 9.1.3 [30] to assess the main effects and interactions among these factors. Tukey’s multiple comparison test was employed to identify significant differences among treatment means, with *p* < 0.05 considered statistically significant.

To ensure the validity of the statistical analysis, the assumptions of normality and homoscedasticity were assessed. Residuals were examined for normality using the Shapiro–Wilk test, and homoscedasticity was evaluated by plotting residuals against fitted values. Where necessary, data transformations were applied to meet these assumptions.

The model utilized in the analysis was as follows:Yijkl = µ + ai + bj + ck + abij + acik + bcjk + abcijk + εijkl
where Yijkl represents the response variable, µ is the overall mean, ai is the effect of HCN source (fresh cassava root or fresh cassava leaves), bj is the effect of HCN level (300 or 600 mg/kg DM), ck is the effect of bacterial treatment (no CUB, *E. faecium* KKU-BF7, or *E. gallinarum* KKU-BC15), and εijkl is the residual error term. Interaction terms were specified as follows: abij: the interaction between HCN source and HCN level, examining how different concentrations of HCN affect responses for each HCN source; acik: the interaction between HCN source and bacterial treatment, assessing how the bacterial treatments interact with HCN sources to influence key outcomes; bcjk: the interaction between HCN level and bacterial treatment, determining how different bacterial treatments interact with HCN concentrations; abcijk: the three-way interaction among HCN source, HCN level, and bacterial treatment, capturing the combined effects of these factors on responses.

## 3. Results

### 3.1. Gas Dynamics and Total Gas Output

The effects of CUB supplementation, along with various HCN sources and levels, on gas kinetics and accumulation are summarized in Table 2, and the cumulative gas production for each substrate treatment is illustrated as gas production curves in Figure 1. No interaction was observed between the HCN source, HCN level, and CUB type on the gas production kinetics, including gas from immediately soluble fractions (a), insoluble fractions (b), the rate of gas production from insoluble fractions (c), the potential total gas production (a + b), and cumulative gas after 96 h (*p* > 0.05). For the main effect of the HCN source, FCR led to significantly higher gas production compared to FCL, with cumulative gas (96 h) values of 203.92 mL and 179.25 mL, respectively (*p* = 0.02). Higher HCN levels (600 mg/kg) resulted in increased cumulative gas production (*p* = 0.02) and gas production from the insoluble fraction (*p* = 0.01) compared to the 300 mg/kg level, with increases of 18% and 16%, respectively. The addition of *E. faecium* KKU-BF7 or *E. gallinarum* KKU-BC15 enhanced cumulative gas production significantly compared to the no-CUB group (*p* = 0.04), with increases ranging from 10.7% to 11.8%.

### 3.2. Ruminal pH and Ammonia-Nitrogen (NH_3_-N) and Cyanide Degradation Efficiency In Vitro

The effects of CUB supplementation with varying sources and levels of HCN on ruminal pH, NH_3_-N concentration, and HCN degradation efficiency in vitro are presented in Table 3. No interactions were observed between HCN sources, levels, and CUB species on ruminal pH and NH_3_-N (*p* > 0.05). However, HCN degradation efficiency showed a significant interaction among these factors. At 4 h of incubation, in vitro rumen pH in the FCR group was lower than in the FCL group (6.65 vs. 6.75; *p* = 0.04). The NH₃-N concentration at 8 h of incubation was highest in the FCL group at 1.03 mg/dL, compared to FCR (*p* = 0.01). CUB supplementation did not change ruminal pH or NH₃-N concentration (*p* > 0.05). Cyanide degradation efficiency was high when FCR was included at 600 mg/kg DM HCN. At 12 h after incubation, HCN degradation efficiency was significantly higher in the *E. gallinarum* KKU-BC15 and *E. faecium* KKU-BF7 groups, reaching 98.44–99.07%. This was significantly greater than in the no-CUB group, which showed a degradation efficiency of 83.20%. 

### 3.3. In Vitro Digestibility

The effects of CUB supplementation, along with different HCN sources and concentrations (300 and 600 mg/kg DM), on IVOMD, IVDMD, IVNDFD, and IVADFD are presented in Table 4. There was no significant interaction between HCN source, HCN level, and CUB type for IVOMD, IVDMD, and IVNDFD (*p* > 0.05). However, the higher HCN level of 600 mg/kg DM significantly increased IVOMD (*p* < 0.01) and IVNDFD (*p* = 0.01) by 4.99% and 7.20%, respectively, compared to the 300 mg/kg DM level. The addition of CUB had no effect on IVOMD, IVDMD, or IVNDFD, except for IVADFD, where a significant interaction among the factors was observed (*p* = 0.02). Specifically, the addition of *E. gallinarum* KKU-BC15 or the high HCN level of 600 mg/kg DM increased IVADFD by 1.42% and 5.53%, respectively, compared to no CUB supplementation or the lower HCN level of 300 mg/kg DM.

### 3.4. Concentration of Volatile Fatty Acids (VFAs)

The effects of cyanide-utilizing bacteria (CUB) supplementation with different HCN sources and levels on VFA concentrations are shown in Table 5. There was no significant interaction between HCN source, HCN level, and CUB type on overall VFA concentrations or profiles (*p* > 0.05). However, compared to the FCL-fed group, FCR supplementation significantly increased total VFA concentration (*p* = 0.03) and propionic acid concentration (*p* = 0.04), raising them from 43.10 to 48.21 mmol/L and 23.48% to 25.06%, respectively. Additionally, a higher HCN level (600 mg/kg DM) increased the propionic acid concentration to 25.17% (*p* = 0.02). The addition of *E. gallinarum* KKU-BC15 further enhanced the propionic acid concentration by 8.97% compared to the group without CUB supplementation (*p* = 0.04).

## 4. Discussion

In this study, experimental concentrate diets were formulated to support optimal growth, health, and milk or meat production in cattle, with crude protein levels between 14 and 18%, as recommended by the NRC [31], to meet these production goals. The rice straw in this experiment contained 2.30% crude protein, and fresh cassava roots provided only 3% crude protein on a dry matter basis, compared to 21.70% in fresh cassava leaves. This variation highlights the limited protein contribution from rice straw and fresh cassava roots, emphasizing the need for protein-rich supplements in the diet to achieve adequate overall crude protein levels for cattle. Cyanide concentrations in fresh cassava leaves (221.37 mg/kg fresh weight) were higher than those in fresh cassava roots (100.60 mg/kg fresh weight). The HCN concentration of Manihot esculenta Kasetsart 50 was found to be 103.5 mg/kg, according to Dagaew et al. [32]. This particular plant was acquired from a local farmer in the Khon Kaen district of Thailand. Prachumchai et al. [11] similarly reported a HCN level of 106.00 mg/kg in fresh cassava roots. Nguyen et al. [9] found that fresh cassava roots had a HCN content of 114 mg/kg. Vetter [33] also reported that HCN levels varied by cassava type, with bitter and sweet varieties containing between 310 and 468 mg/kg. It is important to note that the HCN levels found in our study were slightly lower than those reported in previous studies. This discrepancy may be attributed to various factors, including differences in cassava variety, geographic conditions, and growing practices [9,16]. The variability among cassava varieties and their growing conditions should be acknowledged as potential limitations of this study, as these factors could significantly influence both nutritional content and cyanide levels. Further research is warranted to explore these variabilities and their impact on the nutritional safety of cassava in cattle diets.

The inclusion of concentrate, roughage, and HCN sources (fresh cassava leaves and roots) in the diets led to increased gas production, particularly when CUB were added. This may be due to the availability of starch in fresh cassava roots, which provides a substrate for rumen microbes to generate gas [11]. On the other hand, fresh cassava leaves reduced gas accumulation at 96 h, which may be attributed to their high fiber content (NDF and ADF), potentially limiting digestion and gas production. However, it is important to consider that other factors, such as the presence of anti-nutritional compounds or differences in microbial activity, could also contribute to this effect [34]. Prachumchai et al. [11] observed that increasing the inclusion of fresh cassava root in the diet positively influenced gas production from both the immediately soluble and insoluble fractions, as well as cumulative gas production over a 96 h incubation period. This increase can be attributed to the high starch content in fresh cassava root, which enhances nutrient availability and promotes fermentation processes within the rumen. The starch serves as a readily fermentable carbohydrate, providing energy sources for the ruminal microbes, leading to increased gas production as a by-product of microbial metabolism [32]. Furthermore, the supplementation of CUB is hypothesized to enhance feed digestion and mitigate the toxic effects of HCN by converting HCN into less toxic compounds. This interaction allows for more efficient microbial function and improved fermentation dynamics. Supporting this notion, Sumadong et al. [17] demonstrated that elevated HCN levels negatively impacted gas production from the insoluble fractions and cumulative gas production after 96 h of incubation. However, this detrimental effect was significantly alleviated by CUB supplementation, indicating that these bacteria may aid in maintaining microbial viability and fermentation efficiency in the presence of HCN, ultimately enhancing nutrient utilization and gas production.

In this study, the ruminal pH ranged from 6.30 to 7.17, a level conducive to optimal microbial activity and fiber degradation. This pH range supports bacterial growth, especially for cellulolytic bacteria, which thrive in mildly acidic to neutral conditions, thereby maintaining a balance that allows for the efficient breakdown of fiber [35]. Similar to the findings of Bach et al. [35], ruminal pH levels between 6.5 and 7.0 facilitate fiber digestion without hindering microbial activity. Furthermore, Sombuddee et al. [36] observed that adding CUB to a feed containing HCN helped stabilize pH by enhancing microbial resilience to cyanide stress, thereby creating a favorable environment for fermentation.

The NH_3_-N concentrations observed in this study (19.32–23.43 mg%) indicate adequate nitrogen availability for microbial growth. NH_3_-N is primarily generated through protein hydrolysis and deamination by rumen microbes, serving as an essential nitrogen source for microbial protein synthesis. Adequate NH_3_-N levels ensure the growth of ammonia-utilizing bacteria, which convert NH_3_-N into microbial protein, contributing to the animal’s protein intake [36]. Prachumchai et al. [11] reported that fresh cassava root supplementation increases NH_3_-N concentrations, likely due to the additional fermentable carbohydrates that provide energy for proteolytic bacteria, enhancing NH_3_-N production. Similarly, Cherdthong et al. [3] observed that NH_3_-N levels rose with fresh cassava root supplementation, underscoring its role in balancing energy and nitrogen for microbial growth.

The efficiency of HCN degradation in the rumen was significantly influenced by CUB supplementation. Cyanide-utilizing bacteria metabolize HCN as a nitrogen source, using the rhodanese enzyme to convert cyanide into thiocyanate, a less toxic form that can be safely utilized by rumen microbes [6,11]. This enzymatic degradation of HCN not only detoxifies the rumen environment but also provides nitrogen that can be assimilated into microbial biomass, supporting overall microbial growth and activity.

Regarding in vitro digestibility, fresh cassava roots—high in nonstructural carbohydrates—facilitate rapid microbial fermentation, as these carbohydrates are easily accessible for degradation, thereby promoting bacterial population growth and nutrient utilization [11]. However, increasing fresh cassava leaf levels tended to reduce digestibility. This is likely due to the high fiber content of fresh cassava leaves, which could inhibit microbial accessibility to substrates and slow down the digestion rate [37]. The high fiber in leaves may require more cellulolytic activity, which can be challenging for bacteria in the presence of limited fermentable carbohydrate sources.

The inclusion of CUB appears to support microbial digestion indirectly by reducing feed toxins, such as HCN, thus promoting ammonia production and a favorable microbial environment [4]. Dagaew et al. [32] noted that an increased ratio of fresh cassava roots to rice straw improved fiber digestibility, likely because the starch in cassava root provides additional energy for the microbes involved in fiber degradation. Sumadong et al. [17] observed that sulfur combined with fresh cassava roots enhanced gas production, rumen characteristics, and degradability. This effect is attributed to the synergistic action of nonstructural carbohydrates in cassava roots, which enhance microbial proliferation and activity, leading to improved digestion efficiency. Sombuddee et al. [36] further demonstrated that CUB supplementation increases in vitro dry matter digestibility by 11.16%, likely due to the ability of CUB to reduce toxins and support a microbial environment conducive to optimal fermentation.

Total volatile fatty acids are critical for ruminant energy, providing over 70% of the animal’s energy [14,32]. In this study, supplementation with FCR resulted in significantly greater total VFA and propionic acid concentrations compared to FCL, likely due to its higher energy density, beneficial fermentation characteristics, and reduced toxicity risks from cyanogenic compounds [17]. Cassava roots are particularly rich in carbohydrates, especially starch, which provides an easily fermentable energy source for rumen microorganisms [11]. This high energy availability enhances fermentation activity, leading to an increased VFA production, particularly propionic acid, which is more prominent in starch fermentation. Although cassava leaves have a higher protein content, they also contain more fiber, which can reduce their digestibility and fermentation efficiency in the rumen [19]. Sumadong et al. [17] found that fresh cassava root supplementation significantly affected acetate and butyrate concentrations, with higher total VFA and propionate concentrations but lower acetate and butyrate levels observed. Similarly, Sombuddee et al. [36] reported that potassium HCN and HCN-utilizing bacteria impacted VFA concentrations, with the lowest VFA concentrations occurring at higher HCN levels. This suggests that while HCN levels can initially stimulate certain microbial populations, excessive HCN ultimately hampers VFA production, emphasizing the need for balanced cassava supplementation in ruminant diets [38]. This study additionally revealed that the inclusion of *E. gallinarum* KKU-BC15 increases propionic acid levels by 8.97% compared to the groups lacking CUB supplementation, a result of several interconnected factors. *E. gallinarum* KKU-BC15 is recognized for its capacity to enhance the fermentation process within the rumen. This particular strain effectively utilizes dietary substrates, especially those sourced from cassava, which boosts the production of VFAs, particularly propionic acid [4]. The introduction of CUBs like *E. gallinarum* KKU-BC15 can enhance microbial activity and diversity, which are essential for optimizing the fermentation pathways that favor propionate synthesis [36]. Certain strains of *E. gallinarum* have metabolic functions that facilitate the conversion of substrates into propionic acid; for instance, they can use fumarate as an electron acceptor, transforming it into succinate and then into propionate [4]. This specific metabolic pathway is less active in groups without CUB supplementation, leading to lower concentrations of propionic acid.

## 5. Conclusions

This study highlights the potential of using *E. gallinarum* KKU-BC15 as a valuable feed supplement in diets containing high levels of HCN from fresh cassava root, specifically with 600 mg/kg HCN. The significant increases in gas production, VFA concentration, propionic acid, and in vitro digestibility associated with this supplementation not only suggest improved fermentation efficiency but also point to a promising strategy for mitigating HCN toxicity in ruminants. By optimizing the utilization of cassava as a feed resource, this approach could enhance overall feed efficiency and promote healthier livestock, thereby contributing to sustainable livestock production in regions where cassava is a primary feed source. Further in vivo studies are recommended to explore the potential of *E. gallinarum* KKU-BC15 as a HCN mitigation agent and its application in cattle management.

## Figures and Tables

**Figure 1 animals-14-03269-f001:**
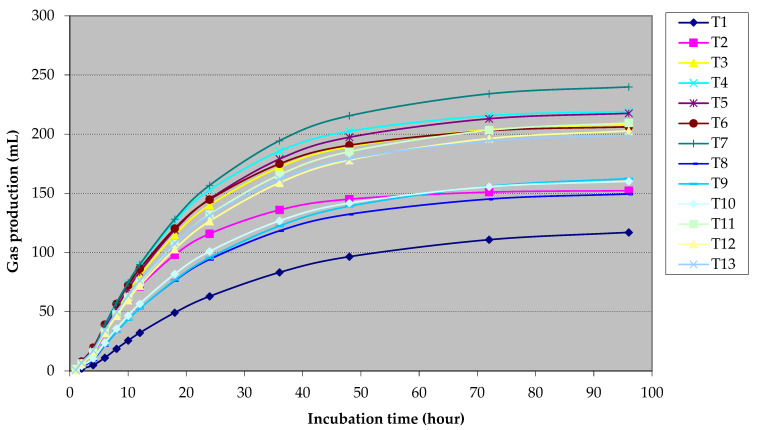
The effect of cyanide-utilizing bacteria (CUB) supplementation with different sources and levels of cyanide on cumulative gas production at different times of incubation. [T1 = control treatment; T2 = fresh cassava root + 300 mg/kg DM of HCN + no CUB; T3 = fresh cassava root + 300 mg/kg DM of HCN + *E. faecium* KKU-BF7; T4 = fresh cassava root + 300 mg/kg DM of HCN + *E. gallinarum* KKU-BC15; T5 = fresh cassava root + 600 mg/kg DM of HCN + no CUB; T6 = fresh cassava root + 600 mg/kg DM of HCN + *E. faecium* KKU-BF7; T7 = fresh cassava root + 300 mg/kg DM of HCN + *E. gallinarum* KKU-BC15; T8 = fresh cassava leaf + 300 mg/kg DM of HCN + no CUB; T9 = fresh cassava leaf + 300 mg/kg DM of HCN + *E. faecium* KKU-BF7; T10 = fresh cassava leaf + 300 mg/kg DM of HCN + *E. gallinarum* KKU-BC15; T11 = fresh cassava leaf + 600 mg/kg DM of HCN + no CUB; T12 = fresh cassava leaf + 600 mg/kg DM of HCN + *E. faecium* KKU-BF7; T13 = fresh cassava leaf + 600 mg/kg DM of HCN + *E. gallinarum* KKU-BC15].

**Table 1 animals-14-03269-t001:** Chemical composition of the substrates used in the in vitro experiment (DM basis, %).

Item	Concentrate, %DM	FCR	FCL	Rice Straw
Ingredients, %DM				
Corn meal	50			
Soybean meal	14			
Fine rice bran	17			
Palm kernel meal	12.5			
Urea	0.5			
Di-calcium phosphate	1.5			
Molasses	3			
Mineral premix	1.5			
Chemical composition				
Dry matter, %	89.61	38.50	26.80	92.50
Organic matter, %DM	94.12	96.87	91.77	89.50
Crude protein, %DM	18.71	3.00	21.70	2.30
Ash, %DM	5.88	3.13	8.23	10.50
Neutral detergent fiber, %DM	52.19	8.00	48.80	71.20
Acid detergent fiber, %DM	22.01	6.90	32.50	44.20
Hydrocyanic acid, ppm	-	100.60	221.37	-

FCR, fresh cassava root; FCL, fresh cassava leaf.

**Table 2 animals-14-03269-t002:** Effect of cyanide-utilizing bacteria (CUB) supplementation with different sources and levels of cyanide on gas kinetics and cumulative gas at 96 h after incubation.

Treatment	Cyanide Source	Level of Cyanide	Type of CUB	Gas Kinetics				Cumulative Gas (mL), 96 h
				a	b	c	|a| + b	
T1	Control			−15.06	136.68	0.03	151.75	113.23
T2	FCR	300	No CUB	−27.37	179.86	0.07	207.23	150.70
T3			*E. faecium* KKU-BF7	−29.82	239.45	0.05	269.27	205.03
T4			*E. gallinarum* KKU-BC15	−34.51	254.92	0.06	289.43	216.20
T5		600	No CUB	−29.76	249.37	0.05	279.14	208.70
T6			*E. faecium* KKU-BF7	−26.62	234.01	0.06	260.63	208.30
T7			*E. gallinarum* KKU-BC15	−29.50	272.11	0.05	301.61	234.57
T8	FCL	300	No CUB	−19.20	170.77	0.04	189.97	144.30
T9			*E. faecium* KKU-BF7	−16.76	183.31	0.04	200.07	161.50
T10			*E. gallinarum* KKU-BC15	−19.73	182.28	0.05	202.02	158.00
T11		600	No CUB	−23.24	235.93	0.04	259.17	209.03
T12			*E. faecium* KKU-BF7	−20.04	226.22	0.05	246.26	203.45
T13			*E. gallinarum* KKU-BC15	−21.97	223.08	0.05	245.05	199.23
SEM				2.21	23.93	0.01	25.60	20.93
Main effect								
HCN source	FCR			−29.60	238.29 ^d^	0.05	267.89 ^d^	203.92 ^d^
	FCL			−20.16	203.60 ^e^	0.04	223.76 ^e^	179.25 ^e^
*p*-value				0.09	0.01	0.71	0.01	0.02
HCN dose	300			−24.57	201.77 ^e^	0.05	226.33 ^d^	172.62 ^e^
	600			−25.49	240.12 ^d^	0.05	265.31 ^e^	210.55 ^d^
*p*-value				0.11	0.01	0.66	<0.01	0.02
CUB specie	No CUB			−24.89	208.98	0.05	233.88	178.18 ^e^
	*E. faecium* KKU-BF7			−23.61	220.75	0.05	244.06	199.57 ^d^
	*E. gallinarum* KKU-BC15			−26.43	233.10	0.05	259.53	202.00 ^d^
*p*-value				0.34	0.09	0.66	0.10	0.04
Interaction								
Control vs. other				0.01	0.01	0.05	0.01	0.01
A × B				0.01	0.01	0.01	0.01	0.01
A × C				0.23	0.08	0.19	0.08	0.08
B × C				0.23	0.08	0.19	0.08	0.08
A × B × C				0.34	0.09	0.66	0.10	0.06

SEM = standard error of the mean. ^d,e^ means in the same column with different superscripts differ (*p* < 0.05). FCR = fresh cassava root; FCL = fresh cassava leaf; a = gas production from the immediately soluble fraction (mL); b = gas production from the insoluble fraction (mL); c = gas production rate constant for the insoluble fraction (mL/h); and a + b = the potential extent of gas production (mL). A = cyanide source (FCR and FCL); B = level of cyanide (300, 600 mg/kg DM); C = type of cyanide-utilizing ruminal bacteria (no CUB, *E. faecium* KKU-BF7, and *E. gallinarum* KKU-BC15). Study interaction effect: A × B is the interaction effect between cyanide source and level of cyanide, A × C is the interaction effect between cyanide source and type of cyanide-utilizing ruminal bacteria, B × C is the interaction effect between level of cyanide 300 and 600 mg/kg DM and type of cyanide-utilizing ruminal bacteria, A × B × C is the interaction effect among cyanide source level of cyanide type of CUB. Significance levels are represented by *p*-value < 0.05.

**Table 3 animals-14-03269-t003:** Effect of cyanide-utilizing bacteria (CUB) supplementation with different sources and levels of pH, ammonia–nitrogen, and cyanide degradation efficiencies.

Treatment	Cyanide Source	Level of Cyanide	Type of CUB	pH		Ammonia–Nitrogen, mg/dL		Cyanide Degradation Efficiency, %		
				4 h	8 h	4 h	8 h	4 h	8 h	12 h
T1	Control			6.92	7.05	22.78	22.22	0 ^g^	0 ^g^	0 ^g^
T2	FCR	300	No CUB	6.72	7.09	22.71	20.39	63.2 ^e^	80.83 ^d^	86.83 ^d^
T3			*E. faecium* KKU-BF7	6.85	7.05	23.04	23.02	75.72 ^c^	90.00 ^b^	100 ^a^
T4			*E. gallinarum* KKU-BC15	6.59	7.17	22.85	19.32	76.44 ^c^	89.81 ^b^	100 ^a^
T5		600	No CUB	6.59	6.97	23.04	19.73	76.14 ^c^	83.52 ^c^	92.69 ^c^
T6			*E. faecium* KKU-BF7	6.58	6.83	23.06	23.21	87.11 ^a,b^	93.85 ^a^	98.70 ^a,b^
T7			*E. gallinarum* KKU-BC15	6.58	6.80	22.05	19.87	87.56 ^a^	93.80 ^a^	100 ^a^
T8	FCL	300	No CUB	6.3	6.71	22.49	22.30	51.20 ^f^	66.27 ^e^	85.61 ^d^
T9			*E. faecium* KKU-BF7	6.71	6.93	22.94	23.13	74.93 ^c^	88.49 ^b^	99.79 ^a^
T10			*E. gallinarum* KKU-BC15	6.68	6.74	22.69	23.43	75.48 ^c^	88.30 ^b^	95.85 ^b,c^
T11		600	No CUB	7.09	7.00	21.53	21.85	67.02 ^d^	78.87 ^d^	92.21 ^c^
T12			*E. faecium* KKU-BF7	6.9	7.08	22.11	19.35	86.45 ^a,b^	93.34 ^a^	97.77 ^a,b^
T13			*E. gallinarum* KKU-BC15	7.02	6.98	22.07	21.64	85.92 ^b^	94.12 ^a^	97.91 ^a,b^
SEM				0.09	0.05	0.29	1.00	0.49	0.80	1.28
Main effect										
HCN source	FCR			6.65 ^b^	6.98 ^a^	22.79 ^a^	20.92 ^b^	77.70 ^a^	88.64 ^a^	95.20 ^a^
	FCL			6.75 ^a^	6.90 ^b^	22.31 ^b^	21.95 ^a^	73.50 ^b^	84.95 ^b^	91.94 ^b^
*p*-value				0.04	0.01	0.01	0.01	<0.01	<0.01	0.01
HCN dose	300			6.64 ^b^	6.95	22.79 ^a^	21.93 ^a^	69.50 ^b^	83.95 ^b^	92.18 ^b^
	600			6.76 ^a^	6.94	22.31 ^b^	20.94 ^b^	81.70 ^a^	89.63 ^a^	94.95 ^a^
*p*-value				0.04	0.75	0.01	0.01	<0.01	<0.01	0.01
CUB specie	No CUB			6.61	6.94	22.44	21.07	64.39 ^b^	77.38 ^b^	83.20 ^b^
	*E. faecium* KKU-BF7			6.76	6.97	22.79	22.17	81.05 ^a^	91.50 ^a^	99.07 ^a^
	*E. gallinarum* KKU-BC15			6.71	6.92	22.42	21.06	81.36 ^a^	91.5 ^a^	98.44 ^a^
*p*-value				0.57	0.06	0.91	1.00	<0.01	<0.01	<0.01
Interaction										
Control vs. other				0.06	0.06	0.47	0.46	<0.01	<0.01	<0.01
A × B				0.06	0.06	0.47	0.09	<0.01	<0.01	<0.01
A × C				0.13	0.51	0.08	0.52	<0.01	<0.01	0.12
B × C				0.13	0.51	0.08	0.52	<0.01	<0.01	0.12
A × B × C				0.57	0.60	0.91	0.99	<0.01	<0.01	<0.01

SEM = standard error of the mean. ^a–g^ means in the same column with different superscripts differ (*p* < 0.05). FCR = fresh cassava root; FCL = fresh cassava leaf. A = cyanide source (FCR and FCL); B = level of cyanide (300 and 600 mg/kg DM); C = type of cyanide-utilizing ruminal bacteria (no CUB, *E. faecium* KKU-BF7, and *E. gallinarum* KKU-BC15). Study interaction effect: A × B is the interaction effect between cyanide source and level of cyanide, A × C is the interaction effect between cyanide source and type of cyanide-utilizing ruminal bacteria, B × C is the interaction effect between level of cyanide 300 and 600 mg/kg DM and type of cyanide-utilizing ruminal bacteria, A × B × C is the interaction effect among cyanide source level of cyanide type of CUB. Significance levels are represented by *p*-value < 0.05.

**Table 4 animals-14-03269-t004:** Effect of cyanide-utilizing bacteria (CUB) supplementation with different sources and level of cyanide on in vitro digestibility.

Treatment	Cyanide Source	Level of Cyanide	Type of CUB	IVOMD, %	IVDMD, %	IVNDFD, %	IVADFD, %
T1	Control			82.59	77.97	47.37	22.38 ^f^
T2	FCR	300	No CUB	83.54	74.26	59.80	23.27 ^f^
T3			*E. faecium* KKU-BF7	83.35	73.75	61.45	21.66 ^f^
T4			*E. gallinarum* KKU-BC15	83.24	72.16	63.43	31.67 ^c,d^
T5		600	No CUB	83.88	77.90	67.16	33.13 ^b,c,d^
T6			*E. faecium* KKU-BF7	84.47	78.80	69.67	27.47 ^e^
T7			*E. gallinarum* KKU-BC15	84.49	79.34	69.10	29.00 ^d,e^
T8	FCL	300	No CUB	86.39	78.62	62.97	30.03 ^d,e^
T9			*E. faecium* KKU-BF7	85.48	76.31	58.80	34.73 ^a,b,c^
T10			*E. gallinarum* KKU-BC15	84.82	75.81	58.56	37.36 ^a^
T11		600	No CUB	87.14	81.75	67.47	33.97 ^b,c,d^
T12			*E. faecium* KKU-BF7	87.09	81.56	67.30	34.75 ^a,b,c^
T13			*E. gallinarum* KKU-BC15	87.05	81.46	67.50	36.31 ^a,b^
SEM				0.49	0.65	0.81	1.27
Main effect							
A	FCR			83.82 ^b^	76.04 ^a^	65.10 ^a^	26.83 ^a^
	FCL			86.33 ^a^	79.25 ^b^	63.77 ^b^	35.77 ^b^
*p*-value				<0.01	<0.01	0.01	<0.01
B	300			84.47 ^b^	75.15 ^b^	60.83 ^b^	28.54 ^b^
	600			85.68 ^a^	80.14 ^a^	68.03 ^a^	34.07 ^a^
*p*-value				<0.01	<0.01	0.01	<0.01
C	No CUB			85.24	78.13	64.35	30.80 ^b^
	*E. faecium* KKU-BF7			85.10	77.6	64.30	30.90 ^b^
	*E. gallinarum* KKU-BC15			84.90	77.19	64.65	32.22 ^a^
*p*-value				0.34	0.06	0.61	0.02
Interaction								
Control vs. other				0.01	<0.01	<0.01	<0.01
A × B				0.01	0.63	<0.01	<0.01
A × C				0.18	0.21	0.01	0.92
B × C				0.18	0.21	0.01	0.92
A × B × C				0.34	0.06	0.61	0.02

SEM = standard error of the mean. ^a–f^ means in the same column with different superscripts differ (*p* < 0.05). FCR = fresh cassava root; FCL = fresh cassava leaf. IVDMD, in vitro digestibility of dry matter; IVOMD, in vitro digestibility of organic matter; IVNDFD, in vitro digestibility of neutral detergent fiber degradability; IVADFD, in vitro digestibility of acid detergent fiber degradability. A = cyanide source (FCR and FCL); B = level of cyanide (300 and 600 mg/kg DM); C = type of cyanide-utilizing ruminal bacteria (no CUB, *E. faecium* KKU-BF7, and *E. gallinarum* KKU-BC15). Study interaction effect: A × B is the interaction effect between cyanide source and level of cyanide, A × C is the interaction effect between cyanide source and type of cyanide-utilizing ruminal bacteria, B × C is the interaction effect between level of cyanide 300 and 600 mg/kg DM and type of cyanide-utilizing ruminal bacteria, A × B × C is the interaction effect among cyanide source level of cyanide type of CUB. Significance levels are represented by *p*-value < 0.05.

**Table 5 animals-14-03269-t005:** Effect of cyanide-utilizing bacteria (CUB) supplementation with different sources and levels of cyanide on volatile fatty acids (VFAs) concentrations.

Treatment	Cyanide Source	Level of Cyanide	Type of CUB	Total VFA, mmol/L	VFA Profiles, %		
					Acetic Acid	Propionic Acid	Butyric Acid
T1	Control			39.22	64.80	25.08	10.12
T2	FCR	300	No CUB	43.04	67.55	23.34	9.11
T3			*E. faecium* KKU-BF7	50.36	65.36	25.41	9.23
T4			*E. gallinarum* KKU-BC15	51.98	66.68	23.21	10.11
T5		600	No CUB	45.84	67.00	23.77	9.23
T6			*E. faecium* KKU-BF7	48.16	64.22	25.96	9.82
T7			*E. gallinarum* KKU-BC15	49.90	62.56	28.67	8.77
T8	FCL	300	No CUB	44.90	67.00	19.88	13.12
T9			*E. faecium* KKU-BF7	46.80	63.77	24.69	11.54
T10			*E. gallinarum* KKU-BC15	40.17	63.45	23.67	12.88
T11		600	No CUB	42.71	65.73	21.82	12.45
T12			*E. faecium* KKU-BF7	44.07	66.14	22.44	11.42
T13			*E. gallinarum* KKU-BC15	39.97	63.63	28.36	8.01
SEM				1.79	1.26	1.60	0.48
Main effect							
HCN source	FCR			48.21 ^a^	65.56	25.06 ^a^	9.38
	FCL			43.10 ^b^	64.95	23.48 ^b^	11.57
*p*-value				0.03	0.08	0.04	0.59
HCN dose	300			46.07	65.54	23.59 ^a^	10.87
	600			45.11	64.88	25.17 ^b^	9.95
*p*-value				0.42	0.08	0.02	0.59
CUB species	No CUB			44.61	66.28	22.93 ^a^	10.79
	*E. faecium* KKU-BF7			46.80	65.15	24.29 ^a,b^	10.56
	*E. gallinarum* KKU-BC15			45.57	64.63	25.19 ^b^	10.19
*p*-value				0.92	0.89	0.04	0.17
Interaction							
Control vs. other				0.01	0.01	<0.01	0.03
A × B				0.01	0.01	<0.01	0.03
A × C				0.02	0.80	0.87	0.75
B × C				0.02	0.80	0.87	0.75
A × B × C				0.92	0.89	0.48	0.17

SEM = standard error of the mean. ^a,b^ means in the same column with different superscripts differ (*p* < 0.05). FCR = fresh cassava root; FCL = fresh cassava leaf. A = cyanide source (FCR and FCL); B = level of cyanide (300 and 600 mg/kg DM); C = type of cyanide-utilizing ruminal bacteria (no CUB, *E. faecium* KKU-BF7, and *E. gallinarum* KKU-BC15). Study interaction effect: A × B is the interaction effect between cyanide source and level of cyanide, A × C is the interaction effect between cyanide source and type of cyanide-utilizing ruminal bacteria, and B × C is the interaction effect between level of cyanide 300 and 600 mg/kg DM and type of cyanide-utilizing ruminal bacteria. Significance levels are represented by *p*-value < 0.05.

## Data Availability

The data that support the findings of this study are available from the corresponding author upon reasonable request.

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
