# Peer review of "Exploring the In Vitro Effects of Cassava Diets and Enterococcus Strains on Rumen Fermentation, Gas Production, and Cyanide Concentrations"

_animals, 2024, doi:10.3390/ani14223269_

Round 1

Reviewer 1 Report

Comments and Suggestions for Authors

Abstract

According to the instructions for authors, the abstract should not exceed 250 words. The abstract presented exceeds the maximum word limit allowed.

Introduction:

L49-52, there is redundancy in the explanation of the advantages of fresh cassava as a food source, particularly when comparing it to cassava chips. The phrases ‘economic efficiency’ and ‘cost-effectiveness’ are repetitive. Also, the expression ‘farmers can grow cassava on their own properties’ is confusing, it should be simpler.

You could explore the issue of cassava toxicity more in the introduction, specifically in which context or contexts cassava poisoning can occur and what the consequences are.

Between paragraph L61 and paragraph L62 there is an abrupt change of topic, and there is no sentence linking toxicity with the detoxification capacity of ruminants.

L62 - 72- They should better explain the role of the enzymes rhodanase and β-mercaptopyruvate sulfurtransferase in detoxification.

Makes reference to the units ‘mV’ and ‘S (/h)’, but does not indicate their meaning in the manuscript.

Material and methods

Why were the cyanide values of 300 and 600 mg/kg DM chosen?

How were the assumptions of normality and homoscedasticity checked?

Results:

- The tables are very long and could be summarised or subdivided or presented in another way.

The accumulated values of gas production over 96 hours should have been graphed, not just the accumulated value over 96 hours. The kinetics study should also have been carried out for several hours, not just the 96 hours of incubation.

Discussion:

- L341-356 - The study does not discuss the potential limitations of the work, such as the variability between different varieties of cassava and the growing conditions.

- Why do they say that fibre from fresh cassava leaves limited gas production? Couldn't it be another factor?

- An explanation of the interaction between HCN and VFA synthesis is lacking.

Conclusion:

The conclusions are too generalised, they should clearly respond to the proposed objective.

General notes:

- There is unnecessary repetition of information in various sections of the document, which can confuse the reader. For example, the description of the treatments and experimental design is repeated in different parts of the text.

- Latin words such as in vitro or in vivo should be italicised throughout the text.

- There are different formatting options throughout the text, including font size and type. Authors should standardise the entire manuscript.

- Sometimes 300 and 600 mg/kg DM appears, sometimes only mg/DM. Everything should be standardised.

- Tables that are too long could be summarised or subdivided.

Author Response

Response to Reviewer 1:

Abstract

According to the instructions for authors, the abstract should not exceed 250 words. The abstract presented exceeds the maximum word limit allowed.

Response: Thank you for recommendation and now the abstract was reduced into 244 words. Also, we have modified regarding to comment from Reviewers 2. Please see in manuscript.

Introduction:

L49-52, there is redundancy in the explanation of the advantages of fresh cassava as a food source, particularly when comparing it to cassava chips. The phrases ‘economic efficiency’ and ‘cost-effectiveness’ are repetitive. Also, the expression ‘farmers can grow cassava on their own properties’ is confusing, it should be simpler.

Response: Thank you for recommendation and we have modified it to “In tropical regions, fresh cassava root and leaves provide an affordable source of energy and protein for ruminants, positioning them as a viable alternative to cassava chips. This accessibility, along with the ease of local cultivation, makes cassava a popular feed choice among farmers” Please see in manuscript.

You could explore the issue of cassava toxicity more in the introduction, specifically in which context or contexts cassava poisoning can occur and what the consequences are.

Response: Thank you for recommendation and we have modified it to “Fresh cassava can be a valuable energy source in ruminant diets, but its use is limited by the presence of hydrocyanic acid (HCN), which poses a risk of prussic acid toxicity or hydrocyanic poisoning in animals [5,6]. Toxicity risks from cassava depend on both the amount consumed and the animal's physiological condition. Poisoning is typically triggered when animals consume excessive quantities, particularly in cases where cas-sava constitutes a large part of the diet or is fed without processing. Ingesting feeds with over 500 mg HCN/kg on a dry-weight basis is considered dangerous, as noted by Aminlari et al. [8]. Fresh cassava roots have been found to contain approximately 114 mg/kg of HCN on a fresh weight basis, which, if unprocessed, can contribute to sig-nificant HCN exposure in livestock diets [9]. Prolonged exposure can impair animal health, manifesting as respiratory distress, reduced growth performance, or even mor-tality in severe cases. Thus, reducing HCN concentrations in cassava feed is critical for ensuring its safe inclusion in ruminant diets.” Please see in manuscript.

Between paragraph L61 and paragraph L62 there is an abrupt change of topic, and there is no sentence linking toxicity with the detoxification capacity of ruminants.

Response: Thank you for recommendation and we have modified it. Please see in manuscript.

L62 - 72- They should better explain the role of the enzymes rhodanase and β-mercaptopyruvate sulfurtransferase in detoxification.

Response: Thank you for recommendation and we have modified it. Please see in manuscript.

Makes reference to the units ‘mV’ and ‘S (/h)’, but does not indicate their meaning in the manuscript.

Response: Thank you for recommendation and we have modified it. Please see in manuscript.

Material and methods

Why were the cyanide values of 300 and 600 mg/kg DM chosen?

Response: The cyanide levels of 300 and 600 mg/kg DM were selected based on prior research conducted by Supapong and Cherdtong (2022; Fermentation 2022, 8, 146), which examined the effects of feeding fresh cassava root containing different HCN concentrations on rumen fermentation, feed utilization, and blood metabolites in Thai native beef cattle. Their study included HCN levels of 0, 300, 450, and 600 mg/kg DM, with rhodanese enzyme added to the concentrated feeds at a rate of 1 mg per 104 ppm HCN. This approach enabled controlled detoxification of HCN without adverse effects on animal health, while also demonstrating improvements in crude protein digestibility, ruminal ammonia-N levels, and blood thiocyanate concentration. By selecting the levels of 300 and 600 mg/kg DM, our study builds upon Supapong and Cherdtong’s findings, specifically targeting the threshold levels that influence HCN detoxification efficacy and the ruminant's capacity to safely tolerate cassava feed. We have described into the manuscript.

How were the assumptions of normality and homoscedasticity checked?

Response: Thank you for recommendation and we have described it in section of “2.6. Statistical analysis” to “To ensure the validity of the statistical analysis, the assumptions of normality and homoscedasticity were assessed. Residuals were examined for normality using the Shapiro-Wilk test, and homoscedasticity was evaluated by plotting residuals against fitted values. Where necessary, data transformations were applied to meet these assumptions”. Please see in manuscript.

Results:

- The tables are very long and could be summarised or subdivided or presented in another way.

Response: Thank you for your feedback and comments regarding the length of the tables. In response, we would like to clarify that the extensive detail in our tables is necessary due to several key factors. First, our study includes 13 distinct treatment groups, which are critical for capturing the full scope of the factorial design and allowing comprehensive comparisons between all treatment combinations. Additionally, the complex factorial structure of our study, characterized by a 2×2×3(+1) arrangement, incorporates three primary factors—HCN source, HCN level, and bacterial treatments—with various levels within each factor. Summarizing the results would limit the clarity of these factorial interactions, which are essential for understanding the independent and interactive effects. Furthermore, each treatment group was analyzed across multiple response variables, including gas production, cyanide reduction, and in vitro digestibility, necessitating detailed data presentation to ensure transparency and enable accurate interpretation of the findings. Finally, the detailed statistical analyses, which include main effects and interaction effects, require comprehensive data in the tables to convey important insights. While we understand the concern about table length, we believe that this detailed format allows for clearer, more comprehensive presentation of the results, ultimately facilitating accurate interpretation by readers. Thank you for your consideration.

The accumulated values of gas production over 96 hours should have been graphed, not just the accumulated value over 96 hours. The kinetics study should also have been carried out for several hours, not just the 96 hours of incubation.

Response: We appreciate your suggestion and we have now provided a graph of gas production over 96 h. Please see in the Figure 1.

Discussion:

- L341-356 - The study does not discuss the potential limitations of the work, such as the variability between different varieties of cassava and the growing conditions.

Response: Thank you for recommendation and we have discussed it. Please see in manuscript.

- Why do they say that fibre from fresh cassava leaves limited gas production? Couldn't it be another factor?

Response: Thank you for recommendation and we have modified it to “On the other hand, fresh cassava leaves reduced gas accumulation at 96 hours, which may be attributed to their high fiber content (NDF and ADF), potentially limiting di-gestion and gas production. However, it is important to consider that other factors, such as the presence of anti-nutritional compounds or differences in microbial activity, could also contribute to this effect [34].” Please see in manuscript.

- An explanation of the interaction between HCN and VFA synthesis is lacking.

Response: Thank you for recommendation and we have modified it to “Total volatile fatty acids (VFAs) are critical for ruminant energy, providing over 70% of the animal's energy [14]. In this study, an increased cassava content in the substrate led to a rise in crude fiber, which may have promoted the synthesis of acetate and butyrate. However, excessive cassava addition inhibited these processes, likely due to elevated levels of HCN produced during cassava fermentation. HCN is toxic to microorganisms, disrupting their metabolic activities and negatively impacting digestion [32]. This inhibition can reduce the overall microbial population responsible for VFA synthesis, resulting in lower acetate and butyrate production. Sumadong et al. [17] found that fresh cassava root supplementation significantly affected acetate and butyrate concentrations, with higher total VFA and propionate concentrations but lower acetate and butyrate levels observed. Similarly, Sombuddee et al. [36] reported that potassium cyanide and cyanide-utilizing bacteria impacted VFA concentrations, with the lowest VFA concentrations occurring at higher cyanide levels. This suggests that while HCN levels can initially stimulate certain microbial populations, excessive HCN ultimately hampers VFA production, emphasizing the need for balanced cassava supplementation in ruminant diets [38].” Please see in manuscript.

Conclusion:

The conclusions are too generalised, they should clearly respond to the proposed objective.

Response: Thank you for recommendation and we have modified it to “Thank you for recommendation and we have modified it to “This study highlights the potential of using E. gallinarum KKU-BC15 as a valuable feed supplement in diets containing high levels of HCN from fresh cassava root, specifically with 600 mg/kg HCN. The significant increases in gas production, VFA concentration, propionic acid, and in vitro digestibility associated with this supplementation not only suggest improved fermentation efficiency but also point to a promising strategy for mitigating HCN toxicity in ruminants. By optimizing the utilization of cassava as a feed resource, this approach could enhance overall feed efficiency and promote healthier livestock, thereby contributing to sustainable livestock production in regions where cassava is a primary feed source. Further in vivo studies are recommended to explore the potential of E. gallinarum KKU-BC15 as a HCN mitigation agent and its application in cattle management” Please see in manuscript.” Please see in manuscript.

General notes:

- There is unnecessary repetition of information in various sections of the document, which can confuse the reader. For example, the description of the treatments and experimental design is repeated in different parts of the text.

Response: Thanks so much. The description of the treatments and experimental design is repeated in “2.6. Statistical analysis” and thus we have removed.

- Latin words such as in vitro or in vivo should be italicised throughout the text.

Response: Thanks so much. However, the guideline of MDPI journal suggest that Latin words such as in vitro or in vivo should not be italicized. Thus, we did not revised as your concern.

- There are different formatting options throughout the text, including font size and type. Authors should standardise the entire manuscript.

Response: Thank you and we have tried our best to fixed it.

- Sometimes 300 and 600 mg/kg DM appears, sometimes only mg/DM. Everything should be standardised.

Response: Thank you and we have changed whole manuscript as mg/kg DM.

- Tables that are too long could be summarised or subdivided.

Response: Thank you and we have provided reason as above.

We appreciate your valuable feedback, and we hope that the above response will meet your requirements and enable you to approve the manuscript for future publication.

Reviewer 2 Report

Comments and Suggestions for Authors

It is an interesting article, however the authors should make some improvements.

It is suggested that the title specifies that it is an in vitro study, the use of "rumen fermentation" generates in the reader the expectation of an in vivo study.

From lines 160 to 164, it is confusing to talk about 5 bottles that only contain ruminal liquid. The controls and the blanks must be defined without a doubt. How were the blanks prepared, how many were there? Immediately after, they mention 3 runs and 3 repetitions that were not defined beforehand.

Lines 168 to 169 mention 84 bottles (3 bottles * 13 treatments * 2 times) which results in 78 bottles; so, what do the other 6 bottles correspond to?

Lines 170 to 172 mention that 130 bottles were used to study the HCN ED (2 per treatment * 13 treatments * 5 times), but they only mention taking samples at 0, 4, 8 and 12 h. Is the other time 24 h? If so, this should be clearly stated.

Lines 174 to 178 mention the laboratory analytical procedures performed on the concentrate, straw, and HCN sources. Include these lines in the section (L148 -149) where the composition of the substrate and the amount placed in the bottles are precisely described.

Lines 178 to 180 mention the times at which gas production was measured. Was this procedure performed on all bottles (5+84+130)?

Tables 2, 3, 4, and 5 should clearly show the separation between the control treatment and those corresponding to the HCN source (FCR, FCL). In the same tables, the results of the interactions of the factors A, B, C are presented. However, nowhere in the tables is it stated that the units presented correspond to significance.

As a suggestion, it is recommended that Table 2, which is the first of the results, use the letters a, b as superscripts to identify the statistical differences. Continue using the letters in an orderly manner for the other tables.

Author Response

Response to Reviewer 2:

It is an interesting article, however the authors should make some improvements.

Response: Thank you for your positive feedback on our article. We appreciate your acknowledgment of the work and the insights you've provided. We are committed to making the necessary improvements to enhance the quality of our manuscript.

It is suggested that the title specifies that it is an in vitro study, the use of "rumen fermentation" generates in the reader the expectation of an in vivo study.

Response: Thank you for recommendation and we have modified it to “Exploring In Vitro Effects of Cassava Diets and Enterococcus Strains on Rumen Fermentation, Gas Production, and Cyanide Concentration” Please see in manuscript.

From lines 160 to 164, it is confusing to talk about 5 bottles that only contain ruminal liquid. The controls and the blanks must be defined without a doubt. How were the blanks prepared, how many were there? Immediately after, they mention 3 runs and 3 repetitions that were not defined beforehand.

Response: Thank you for recommendation and we have modified it to “The cumulative assessment of gas output was conducted through three distinct runs, each consisting of three replicates [(3 bottles per treatment × 13 treatments) + 5 bottles of blank)] × 3 runs. To establish baseline measurements, a total of five control bottles containing only ruminal inoculum were used as blanks. These blanks were prepared in the same manner as the experimental bottles but without the addition of any substrates. The mean gas output values from these control bottles served as a reference point. The net gas output for each experimental bottle was determined by subtracting the mean values of the blanks from the corresponding measured values.” Please see in manuscript.

Lines 168 to 169 mention 84 bottles (3 bottles * 13 treatments * 2 times) which results in 78 bottles; so, what do the other 6 bottles correspond to?

Response: We apologies for missed take. It is 78 bottles and we have modified. Thank you.

Lines 170 to 172 mention that 130 bottles were used to study the HCN ED (2 per treatment * 13 treatments * 5 times), but they only mention taking samples at 0, 4, 8 and 12 h. Is the other time 24 h? If so, this should be clearly stated.

Response: We apologies for missed take. We have modified by adding other 24 h into manuscript. Thank you.

Lines 174 to 178 mention the laboratory analytical procedures performed on the concentrate, straw, and HCN sources. Include these lines in the section (L148 -149) where the composition of the substrate and the amount placed in the bottles are precisely described.

Response: Thank you for recommendation and we have modified it. Please see in manuscript.

Lines 178 to 180 mention the times at which gas production was measured. Was this procedure performed on all bottles (5+84+130)?

Response: We have indicated above “…The cumulative assessment of gas output was conducted through three distinct runs, each consisting of three replicates [(3 bottles per treatment × 13 treatments) + 5 bottles of blank)] × 3 runs.”. Thank you.

Tables 2, 3, 4, and 5 should clearly show the separation between the control treatment and those corresponding to the HCN source (FCR, FCL). In the same tables, the results of the interactions of the factors A, B, C are presented. However, nowhere in the tables is it stated that the units presented correspond to significance.

Response: Thank you for recommendation. We have separation between control (T1) and HCN treatments (T2-T13) by inserted a line in the table to make it easier for readers to identify the control group. The units of significance have been provided at footnote as “Significance levels are represented by P-value <0.05.” Please see in Tables 2-5.

As a suggestion, it is recommended that Table 2, which is the first of the results, use the letters a, b as superscripts to identify the statistical differences. Continue using the letters in an orderly manner for the other tables.

Response: Thank you for your valuable suggestion regarding the use of superscript letters to indicate statistical differences in Table 2 and subsequent tables. We appreciate your emphasis on clarity in presenting statistical significance. However, we would like to clarify that the letters a, b, and c are already designated in our study to represent kinetic gas measurements. Using these letters for both purposes could lead to confusion and misinterpretation for readers. To maintain clarity and avoid any duplication, we propose using a new series of superscript letters starting from d onward to indicate statistical significance in our results. This approach will ensure that our readers can easily differentiate between kinetic gas measurements and statistical differences without misunderstanding. We believe this solution will enhance the overall clarity of our tables.

We appreciate your valuable feedback, and we hope that the above response will meet your requirements and enable you to approve the manuscript for future publication.

Reviewer 3 Report

Comments and Suggestions for Authors

Dear authors, respectfully, it seems to me that this is a good manuscript. However, some points affect the quality of the manuscript. The main factor that affects the manuscript is the generic writing style that decreases the relevance of the text.

Simple Summary: In the simple summary I am only missing the most important highlighted result.

Lines 17-18: Remove the text “as assessed by gas production kinetics”

Abstract:  My suggestion is to add numbers to improve your description because describing "increase or decrease" is generic. Describe the results in a way other than the simple transcription of tables and figures.

Line 28: Add the experimental units and how the statistical analysis was evaluated.

Lines 43-44: The conclusion is not supported by the results described in the abstract.

Keyword: Reorder the keywords alphabetically.

Introduction: The introduction is ok; however, some parts are very generic. My suggestion is to add numbers to improve your description. Also, avoid informal text description.

Lines 58-61: Which species are these concentrations negative? If possible, please add the limit concentration for the main ruminant species.

Line 79: How much is “low”? Avoid writing generically. Add numbers.

Material and methods

Line 167: According to what methodologies.

Lines 227-235: Factors were used and appear in the statistical equation; however, the description does not show specific interactions regarding the factors.

Line 234: Words like “in vitro” should be described in italics.

Results: The description of the results is very generic, a direct translation from the tables to the text. To improve this description, it is necessary to describe it in another way using numbers. For example: Bacteria A promoted 15% less ammonia production in vitro compared to bacteria B. Bacteria C was similar to the others.

Discussion: The discussion should focus on explaining how the results were obtained. For this, add theories, hypotheses or statements about how you obtained your results, whether biologically, metabolically, physiologically, environmentally, etc. In the current situation, the discussion is a good general review and comparison of data with other authors; however, you need to make a specific description of how your results were obtained. Improve it.

Lines 341-342: What is “optimal production”?

Lines 343-344: Why is this information relevant?

Conclusion: The results in the current format are a description of the results. Please try rewriting the conclusion in a different way.

Author Response

Response to Reviewer 3:

Dear authors, respectfully, it seems to me that this is a good manuscript. However, some points affect the quality of the manuscript. The main factor that affects the manuscript is the generic writing style that decreases the relevance of the text.

Response: Thank you for your thoughtful and constructive feedback. We appreciate your positive assessment of our manuscript and are grateful for your insights regarding areas for improvement. We understand that our writing style may have appeared somewhat generic, which could impact the clarity and relevance of the manuscript. To address this, we have revised the text to enhance specificity, aiming to clearly highlight the study's contributions and relevance to the field. We have made particular efforts to provide detailed explanations of key findings and to ensure that each section accurately reflects the study's objectives and implications. We hope these revisions meet your expectations and improve the overall quality of the manuscript.

Thank you once again for your valuable comments

Simple Summary: In the simple summary I am only missing the most important highlighted result.

Response: Thank you for recommendation and we have modified it. Please see in manuscript.

Lines 17-18: Remove the text “as assessed by gas production kinetics”

Response: Thank you for recommendation and we have modified it. Please see in manuscript.

Abstract:  My suggestion is to add numbers to improve your description because describing "increase or decrease" is generic. Describe the results in a way other than the simple transcription of tables and figures.

Response: Thank you for recommendation and we have modified it. We also revised regarding to Reviewer 1 suggestion to limited word count lower than 250 words. Please see in manuscript.

Line 28: Add the experimental units and how the statistical analysis was evaluated.

Response: We have indicated as “Statistical analysis was performed using the PROC GLM procedure in SAS.”. Please see in abstract.

Lines 43-44: The conclusion is not supported by the results described in the abstract.

Response: Thank you for recommendation and we have modified it. Please see in manuscript.

Keyword: Reorder the keywords alphabetically.

Response: Thank you for recommendation and we have modified it. Please see in manuscript.

Introduction: The introduction is ok; however, some parts are very generic. My suggestion is to add numbers to improve your description. Also, avoid informal text description.

Response: Thank you for recommendation and we have modified it. Please see in manuscript.

Lines 58-61: Which species are these concentrations negative? If possible, please add the limit concentration for the main ruminant species.

Response: Thank you for recommendation and we have modified it. Please see in manuscript.

Line 79: How much is “low”? Avoid writing generically. Add numbers.

Response: We have change to “A rumen pH below 6.0, typically associated with high-concentrate diets, can impair microbial activity and slow HCN degradation [14].” Please see in manuscript.

Material and methods

Line 167: According to what methodologies.

Response: Thank you. The method VFA analysis has been provided in sup-topic of “2.5. Analysis of chemical composition in diets” which indicated VFA measured by “Additionally, the VFA content was analyzed using gas chromatography. The gas chromatograph used for this analysis was the Wilmington, DE 5890A Series II model, and a glass column measuring 180 cm in length and 4 mm in diameter was employed. A combination of 100 g/L SP-1200 and 10 g/L H3PO4 were packed into the column using 80/100 mesh Chromosorb WAW, which was provided by Supelco, a company based in Bellefonte, Pennsylvania, USA.”  Please see in manuscript.

Lines 227-235: Factors were used and appear in the statistical equation; however, the description does not show specific interactions regarding the factors.

Response: Thank for suggestion and we have modified as “The model utilized in the analysis was as follows:

Yijkl = µ + ai + bj + ck + abij + acik + bcjk + abcijk + εijkl

where Yijkl represents the response variable, µ is the overall mean, ai is the effect of HCN source (fresh cassava root or fresh cassava leaves), bj is the effect of HCN level (300 or 600 mg/kg DM), ck is the effect of bacterial treatment (no CUB, E. faecium KKU-BF7, or E. gallinarum KKU-BC15), and εijkl is the residual error term. Interaction terms were specified as follows: abij: the interaction between HCN source and HCN level, examining how different concentrations of HCN affect responses for each HCN source, acik: the interaction between HCN source and bacterial treatment, assessing how the bacterial treatments interact with HCN sources to influence key outcomes, bcjk: the interaction between HCN level and bacterial treatment, determining how different bacterial treatments interact with HCN concentrations, abcijk: the three-way interaction among HCN source, HCN level, and bacterial treatment, capturing the combined effects of these factors on responses.” Please see in the manuscript.

Line 234: Words like “in vitro” should be described in italics.

Response: Thank for comment. However, MDPI policy they prefer no italic for Latin language.

Results: The description of the results is very generic, a direct translation from the tables to the text. To improve this description, it is necessary to describe it in another way using numbers. For example: Bacteria A promoted 15% less ammonia production in vitro compared to bacteria B. Bacteria C was similar to the others.

Response: Thank you for your valuable feedback. We appreciate your suggestion to provide a more detailed and engaging description of the results. We have revised the Results section to include specific numerical comparisons, as recommended. For instance, we now highlight how each bacterial treatment influenced ammonia production, digestibility, and gas kinetics with precise percentage changes and comparative phrases. This revised approach should enhance clarity and improve the flow of information beyond a simple translation of the tables.

Discussion: The discussion should focus on explaining how the results were obtained. For this, add theories, hypotheses or statements about how you obtained your results, whether biologically, metabolically, physiologically, environmentally, etc. In the current situation, the discussion is a good general review and comparison of data with other authors; however, you need to make a specific description of how your results were obtained. Improve it.

Response: Thank you for your valuable feedback. We agree with your suggestion to provide a more in-depth explanation of how our results were obtained. In response, we have revised the discussion to include theories, hypotheses, and mechanisms, focusing on the biological, metabolic, and physiological processes that likely contributed to our findings. This revision aims to clarify the underlying factors influencing our results and provide a more comprehensive understanding beyond data comparison. We appreciate your guidance in strengthening this section.

Lines 341-342: What is “optimal production”?

Response: Thank you and we have modified as “The experimental concentrate diets were formulated to support optimal growth, health, and milk or meat production in cattle. According to NRC [31], crude protein levels between 14-18% are considered adequate for meeting the nutritional requirements necessary to sustain these production levels in cattle.” Please see in the text.

Lines 343-344: Why is this information relevant?

Response: Thank you and we have modified as “The rice straw in this experiment contained 2.30% crude protein, and fresh cassava roots provided only 3% crude protein on a dry matter basis, compared to 21.70% in fresh cassava leaves. This variation highlights the limited protein contribution from rice straw and fresh cassava roots, emphasizing the need for protein-rich supplements in the diet to achieve adequate overall crude protein levels for cattle.” Please see in the text.

Conclusion: The results in the current format are a description of the results. Please try rewriting the conclusion in a different way.

Response: Thank you for recommendation and we have modified it to “This study highlights the potential of using E. gallinarum KKU-BC15 as a valuable feed supplement in diets containing high levels of HCN from fresh cassava root, specifically with 600 mg/kg HCN. The significant increases in gas production, VFA concentration, propionic acid, and in vitro digestibility associated with this supplementation not only suggest improved fermentation efficiency but also point to a promising strategy for mitigating HCN toxicity in ruminants. By optimizing the utilization of cassava as a feed resource, this approach could enhance overall feed efficiency and promote healthier livestock, thereby contributing to sustainable livestock production in regions where cassava is a primary feed source. Further in vivo studies are recommended to explore the potential of E. gallinarum KKU-BC15 as a HCN mitigation agent and its application in cattle management” Please see in manuscript.

We appreciate your valuable feedback, and we hope that the above response will meet your requirements and enable you to approve the manuscript for future publication.

Round 2

Reviewer 1 Report

Comments and Suggestions for Authors

The authors have taken on board all the comments made by the reviewers. 

Much of the manuscript has been completely rewritten, incorporating the suggestions made.

Reviewer 3 Report

Comments and Suggestions for Authors

Dear authors, reviewers make suggestions with the aim of improving the manuscript; however, it is the responsibility of the authors to accept them in full, accept them partially or reject them. I am satisfied with the responses.